# Rethinking Bayesian Optimization with Gaussian Processes: Insights from Hyperspectral Trait Search

**Ruhana Azam**
University of Illinois Urbana-Champaign
razam2@illinois.edu

**Samuel B. Fernandes**
University of Arkansas
samuelbf@uark.edu

**Andrew D.B. Leakey**
University of Illinois Urbana-Champaign
melkebir@illinois.edu

**Alexander E. Lipka**
University of Illinois Urbana-Champaign
alipka@illinois.edu

**Mohammed El-Kebir**
University of Illinois Urbana-Champaign
melkebir@illinois.edu

**Sanmi Koyejo**
Stanford University
sanmi@stanford.edu

## Abstract

The application of Bayesian Optimization using Gaussian Processes (BO-GP) for global optimization problems is ubiquitous across scientific disciplines because, beyond good performance, it supports exact inference, is interpretable, and has straightforward uncertainty quantification. In this paper, we revisit the biological application of BO-GP in searching trait spaces for genomic prediction, which uses genome-wide marker information to predict breeding values for agronomically important traits. Genomic predictions help breeders select desirable plants earlier in the field season without waiting to observe traits later. While these search spaces are known to be sharp and aperiodic, BO-GP is considered a feasible approach. However, our work finds that a simple random search surprisingly achieves comparable performance to BO-GP while requiring significantly less computing cost. Through a careful investigation, we can explain this observation as a limitation of BO-GP using radial basis function kernels (RBF) for sharp and aperiodic functions – where the incompatible structure results in samples similar to random search but with higher computational cost. Our results highlight a blind spot in the current use of BO-GP for scientific applications, such as trait prediction, with sharp and aperiodic search spaces.

## 1 Introduction

Bayesian optimization (BO) is an approach to finding the global optima of an unknown function in situations where the objective function is expensive to evaluate. In Bayesian optimization, a surrogate model is built as the posterior predictive distribution over the full function space. This distribution, which includes a measure of uncertainty, is then used in conjunction with the acquisition function to trade off exploration and exploitation for the next best sample [Fra18]. Historically, GP surrogates functioned well in a wide variety of Bayesian optimization applications, including hyperparameter tuning of machine learning models [SLA12], material design[KOTM16, BXT+16], and drug discovery[Neg11, SSL+21]. Additionally, Gaussian processes have consistently proven to work well in a variety of benchmarks commonly performed on Bayesian optimization methods [LRW23]. We note that while many surrogate functions have been proposed [WHSX16, WI20, LBN+18, IVHW21],

Gaussian processes (GPs) with Matérn kernels are the default choice of surrogate function in BO [RW06].

Although Gaussian processes have proven to be successful in a wide range of problems, their default use sometimes ignores that this performance depends on the characteristics of the search space. In this paper, we revisit the biological application of BO-GP in searching trait spaces for genomic prediction, which uses genome-wide marker information to predict breeding values for agronomically important traits. Genomic predictions help breeders select desirable plants earlier in the field season without waiting to observe traits later. While these search spaces are known to be sharp and aperiodic, BO-GP is considered a feasible approach. However, our work finds that a simple random search surprisingly achieves comparable performance to BO-GP while requiring significantly less computing cost. **Our main contribution** is investigating this observation as a fundamental limitation of BO-GP for sharp and aperiodic functions – where the incompatible structure results in samples similar to random search but with higher computational cost. Our results highlight a blind spot in the current use of BO-GP for scientific applications, such as trait prediction, with sharp and aperiodic search spaces.

## 2    Co-Heritability Search

In this section, we introduce a novel scientific application of Bayesian optimization for co-heritability search and present results comparing the performance of GP-based Bayesian optimization versus random search on this problem.

### 2.1    The Motivation: What is Heritability?

Heritability is defined as the portion of population variance of a trait explained by genetic factors. In other words, it quantifies the degree to which the variance of a trait is attributable to genetic differences versus environmental factors. Traits with high heritability can have their genetic components predicted with greater reliability.

For plant breeders, highly heritable traits are desirable because they can be more reliably controlled during breeding. Collecting data on traits that are known to be desirable (e.g., nitrogen area and specific leaf area) for a batch of plants is a slow and costly procedure. To relieve this issue, recent work shows that certain low-cost traits, like wavelength reflectance, can be correlated with desirable traits (like nitrogen area, specific leaf area), which are known to improve crop yield. Co-heritability is a measure that combines the heritability of the desired trait, the low-cost trait, and the correlation between the two traits. Utilizing low-cost traits that have high co-heritability with desired trait values has been shown to improve the prediction accuracy of the desired trait while reducing the cost of data collection on the desired trait for a batch of crops [FAP+23].

Heritability and co-heritability are estimated by fitting linear mixed effects models (LMM) on a combination of environmental and genetic traits of crops. For each trait in question, a separate LMM model is trained [BMBW15, PM07]. Depending on the size of the trait space, this procedure can quickly become expensive, thus leaving Bayesian optimization methods a choice for conducting a global search of the trait space. Futher details can be found in Appendix B.

### 2.2    Our Co-Heritability Search Space

Our dataset consists of 869 Sorghum Lines from two growouts near the University of Illinois-Urbana Champaign. These data contain two different high-cost, target traits for plant breeders: 1) Nitrogen Area (Narea), and 2) Specific Leaf Area (SLA). Additionally, the data includes hyperspectral reflectance, which is low-cost to collect. The hyperspectral reflectance is used to compute wavelength ratios $(w_1/w_2)$ – known to be correlated with desired traits in plants [RRW+18, LZGJ18]. The goal is to discover the wavelength ratio that exhibits high co-heritability with the desired trait, which, in turn, increases the efficiency of of genomic prediction. Each wavelength is in the range [350nm-2500nm]. A visual representation of co-heritabilities over different wavelength ratios is presented in Figure 1.

Figure 1 shows that the co-heritability search spaces have many modes – specifically, the function has large smooth regions with little signal toward the local peaks. Illustrated in terms of Euclidean distance, we find that the top 1% of points (i.e., peaks or modes) are close to points with the lowest

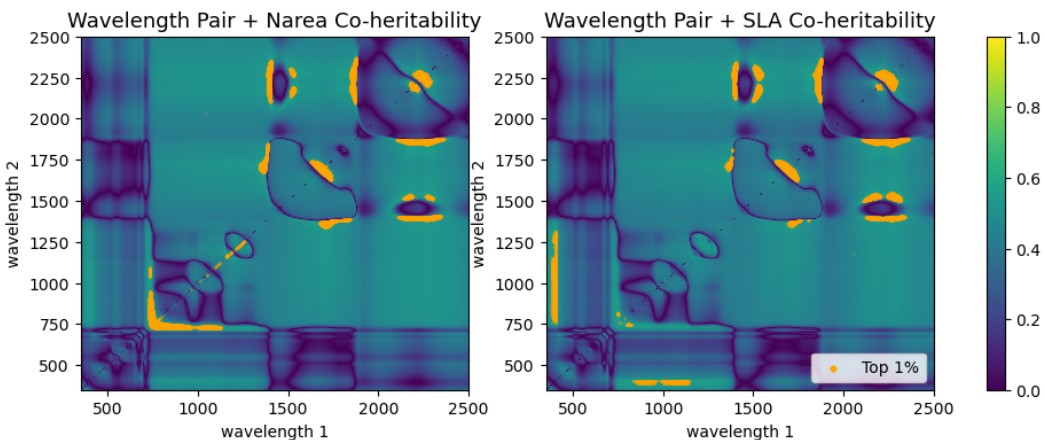

Figure 1: An Illustration of the search space of co-heritabilities of nitrogen area (left) and specific leaf area (right) over at different wavelength ratio. Each point is the computed co-heritability of $\frac{wavelength_1}{wavelength_2}$. Our goal is to estimate the models of this function, i.e., points with high co-heritability

heritability values (i.e., troughs or flat regions), making these modes extremely sharp. Moreover, these patterns are repeating yet not periodic.

In this work, we seek search methods for computing highly heritable traits (i.e., the modes or peaks) without exhaustive enumeration. Note that computing the co-heritability for each trait (in Figure 1, wavelength ratio) is computationally expensive and is not feasible for complex trait spaces. For instance, common trait constructions may be constructed as of ratios of linear combinations of functions, e.g., $\frac{w_1+w_2}{w_3+w_4}$), which results in a 4-dimensional space – so very quickly, the number of points overwhelms any reasonable computational infrastructure. Nevertheless, domain experts (along with experimental evidence) expect that characteristics shown in the lower dimensional cases will persist when the problem's dimensions are increased. Our work will focus on the two-dimensional trait spaces because it is already a useful trait construction, it is computationally non-trivial, and its clear structure is for illustrative spaces.

## 2.3 An Empirical Study: BO-GP For Co-Heritability Search

In this section, we empirically evaluate Bayesian optimization search on two different co-heritability spaces: 1) wavelength ratios & nitrogen area, and 2) wavelength ratios & specific leaf area. We used Gaussian processes with a Matérn ($\nu = 5/2$) kernel as the surrogate model, as suggested by domain experts and common practice. The experiments were run with five seeds for two acquisition functions: 1) Expected Improvement (EI) and 2) Upper Confidence Bound (UCB). A uniform random search was used as a simple baseline.

In Figures 2, BO-UCB was able to detect the top 1% of the search space the fastest – finding the highest co-heritability after 310 iterations. Across the board, random search outperformed BO-EI. Although BO-UCB outperformed random search, it was only by a small margin. Considering that the runtime of random search is $O(1)$ per iteration while fitting a Gaussian process takes $O(n^2)$ for the nth iteration, it is clear that the cost-performance trade-off does not justify using GP-based Bayesian optimization despite its slightly better performance. In practice, the trade-off curves can be seen in Figure 3.

**Main Claim:** Our experiments show that using Gaussian processes-based Bayesian optimization for finding high co-heritabilities may not perform much better than random search. We find several characteristics leading to the small gap in performance between GP-based Bayesian optimization and random search. First is **sharpness** – there are large smooth regions with little to no signal towards the optimal co-heritability. Regarding Euclidean distance, the top 1% of points are extremely close to points with the lowest heritability values, making these regions extremely sharp. Since Gaussian processes with Matérn kernels are most effective for representing smooth functions, they are unlikely to represent extremely sharp regions accurately. Second is **aperiodicity**, the aforementioned

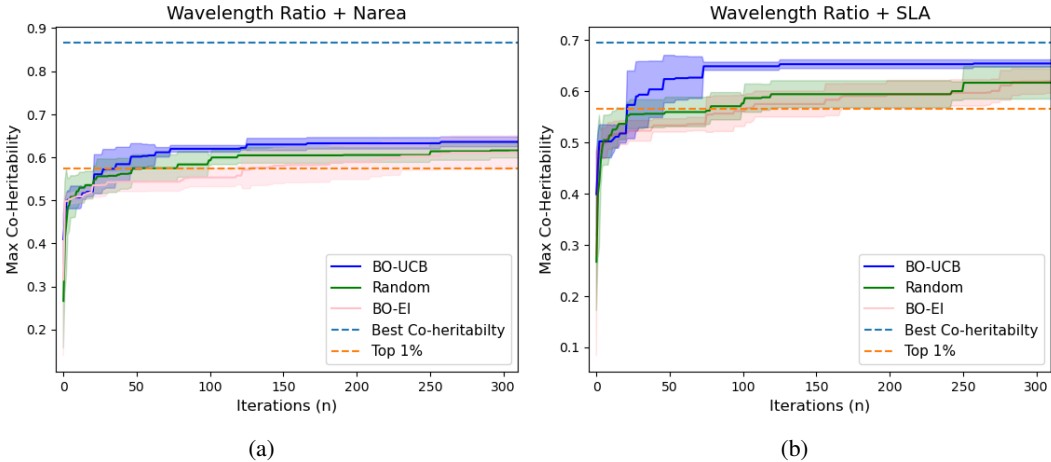

(a)                       (b)

Figure 2: Performance (max co-heritability) vs. number of iterations for Gaussian Process Bayesian optimization for two different co-heritabilities spaces 2a nitrogen area 2b specific leaf area.

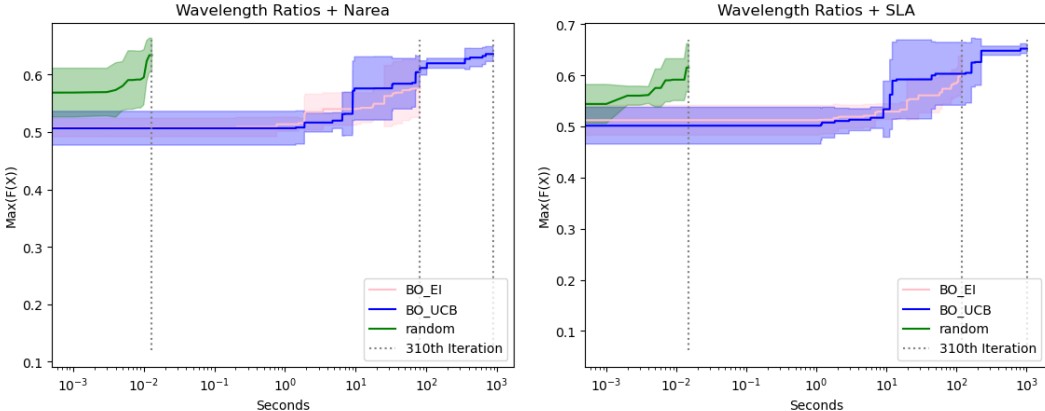

Figure 3: Performance (mean and standard deviation bands) vs. time to search over two co-heritability settings, Narea & Wavelength Ratios (left) and SLA & Wavelength Ratios (right). Each method is run for 310 iterations.

patterns are repeated throughout the search space, yet in a non-periodic manner. This implies that a different choice of kernel cannot easily alleviate the sharpness issue. We further note that any additional hyperparameter search adds additional computational overhead – despite only limited gains in performance.

## 3   Investigating the limitations of Gaussian Processes: Sharp Functions

To further investigate the limitations of Gaussian processes with Matérn kernels, we investigate how the sharpness of a search space affects the performance of BO-GP in a toy settings.

### 3.1   Toy Setting: Gaussian Distributions

Consider a 1D search over a Gaussian distribution with different variances. Since the search space is unimodal, the local and global levels of smoothness for the search space are fully controlled by the variance.

Figure 4a shows three different Gaussian functions with three different levels of sharpness (from a wide Gaussian to approaching a delta function). Similar to Section 2.3, we evaluate the performance

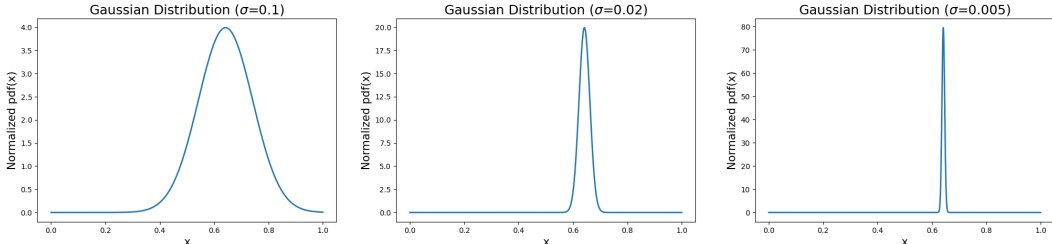

(a) Three Gaussian distributions with varying standard deviations to mimic different sharpness levels. The probability density of x (pdf) are normalized to simplify comparison.

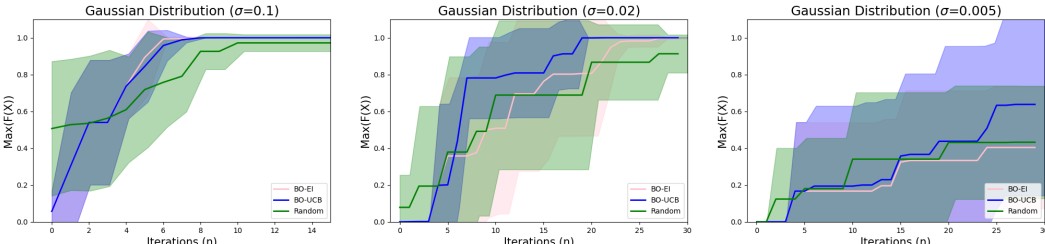

(b) Performance (mean and standard deviation bands) vs. number of iterations for the Gaussian distributions. The results are arranged from the smoothest search space (left) to the sharpest search space (right).

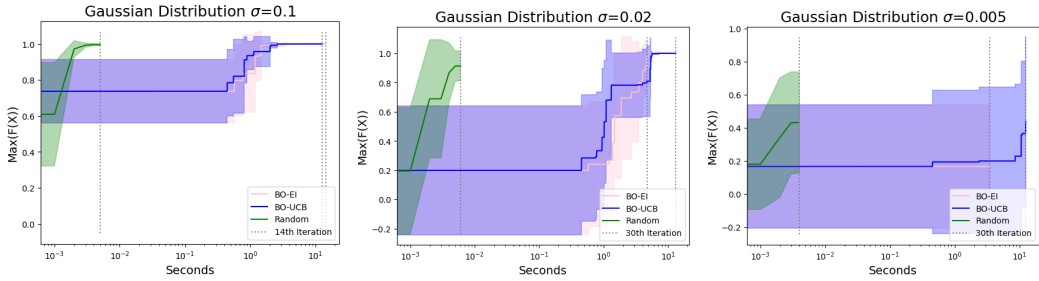

(c) Performance (mean and standard deviation bands) vs. time to search over three Gaussian distributions. The results are arranged from the smoothest search space (left) to the sharpest search space (right). Each method is run for the number of iterations matching in Figure 4b

of GP Matérn kernel based Bayesian optimization with the acquisition functions EI and UCB. Random search was evaluated as a baseline method.

In the smoothest setting, Figure 4b shows that the BO methods can find the optimal point in fewer iterations than random search. However, the variances are large for the sharpest setting, and the maximum point found is significantly lower across the board. We see that BO-EI performs similarly to random search while BO-UCB outperforms both. Yet UCB displays very high variance in performance. We see that utilizing an acquisition function like UCB can improve search performance in a sharp setting. Again, when considering the computational complexity of fitting GPs, as seen in Figure 4c, the performance improvements of GPs may be considered marginal. We further illustrate this issue using benchmarks in the next section – particularly focused on other standardized canonical sharp and aperiodic search spaces.

## 3.2 Additional Benchmarks

In this section, we introduce two functions that have characteristics of difficult search spaces: sharpness and aperiodicity.

**Michalewicz:** is a function often used to benchmark evolutionary algorithms [Mic96]. The function is typically expressed as a multi-dimensional problem and is characterized by its non-linearity, multimodality, and rugged landscape, making it a challenging test case for optimization algorithms. The two-dimensional case of this function, shown in Figure 5a, has sharp and aperiodic peaks.

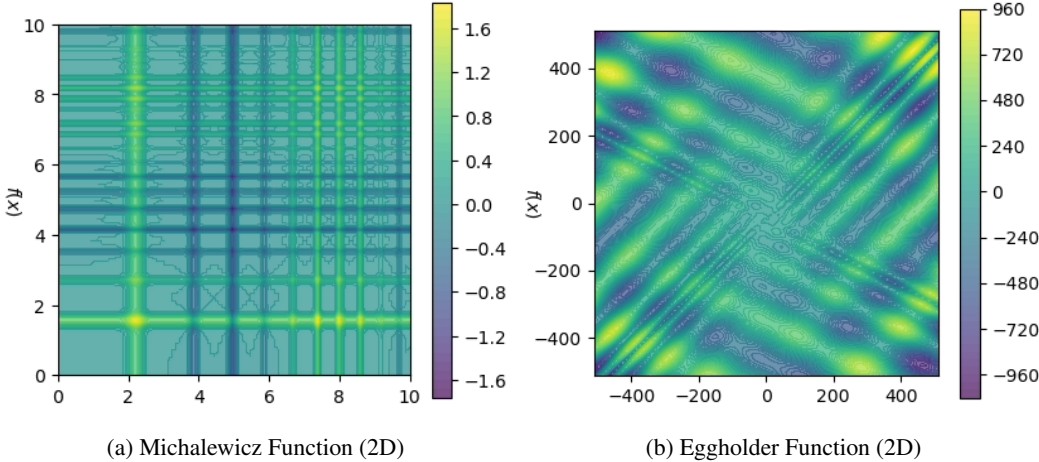

(a) Michalewicz Function (2D)    (b) Eggholder Function (2D)

Figure 5: Two-dimensional multi-modal functions which will be used as additional benchmarks.

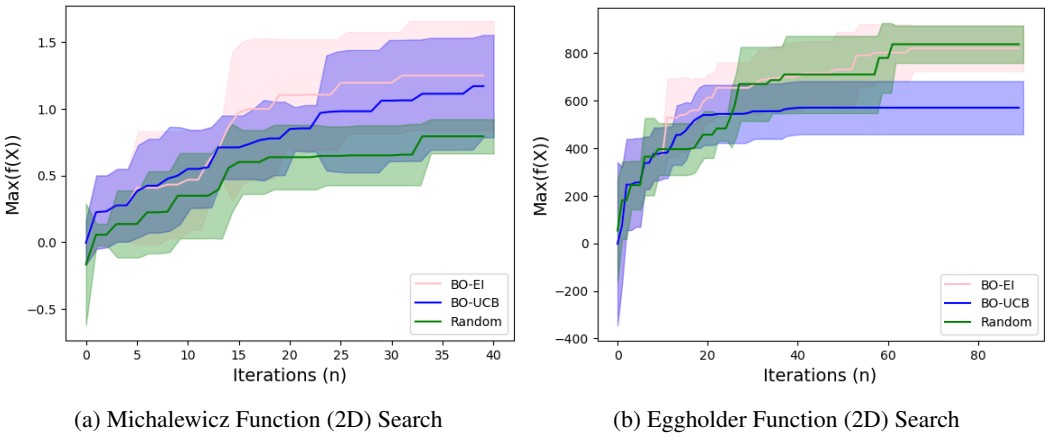

(a) Michalewicz Function (2D) Search    (b) Eggholder Function (2D) Search

Figure 6: BO-GP performance across iterations on two synthetic functions.

**Eggholder:** is a two-dimensional function, also commonly used to benchmark evolutionary algorithms [WRDM96]. It features a distinctive landscape with multiple peaks and valleys, resembling an egg holder. As seen in Figure 5b, the function contains many small and large peaks at varying levels of sharpness.

### 3.2.1 Results

In this section, we benchmark Matérn kernel based BO-GP with two acquisition functions Expected Improvement (EI) and Upper Confidence Bound(UCB) on the the aforementioned setting.

As seen in Figure 6b, random search marginally outperforms BO-EI, while significantly outperforming BO-UCB. Similarly, benchmarking on the Michalewicz function (Figure 6a) shows that BO-GP performs better than random search. However, the improvement is marginal given the high variance of the BO-GP results. Additionally, considering that BO-GP has a runtime complexity of $O(n^2)$ for the nth sample, versus $O(1)$ for random search, the small performance boost comes at a disproportionately high computational cost. The substantial difference in runtime between BO-GP and random search is evident in Figure 7. Overall, random search provides a better performance vs. computational cost trade-off on these sharp, aperiodic benchmarks.

This can be explained for two reasons, 1) the length scales of the global optimas are small, so it hard to find any signal of where the global optima could be 2) the Gaussian process posterior cannot accurately depict the underlying function.

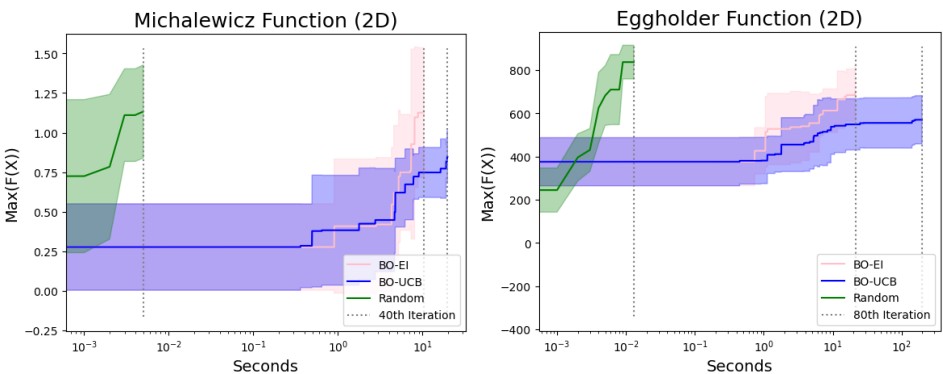

Figure 7: BO-GP performance based on run time for two synthetic functions, Michalewicz (left) and Eggholder (right). Each method was run the same number of iterations as Figure 6.

## 4 Discussion

Gaussian processes with Matérn kernels are widely used as the default choice of surrogate function for Bayesian optimization methods. Although they are known to be effective in many scientific applications and benchmarks, their performance strongly depends on the characteristics of the search space. This paper considers the scientific application of co-heritability, which requires a search over wavelength ratios. Our results show that Bayesian optimization using these surrogates performs comparably to random search. We explain this as the expected behavior for sharp and aperiodic search spaces, such as this, which makes it difficult for Gaussian processes with Matérn kernels to capture meaningful structure. These characteristics are uncommon in the most popularly utilized benchmark settings for Bayesian optimization. We leave co-heritability search as a novel scientific application where the search space has features that are difficult for BO-GP with Matérn kernels to navigate.

## 5 Future Work

Our results highlight fundamental limitations of Gaussian processes for sharp, aperiodic functions like those found in co-heritability search spaces. We propose three promising directions to address these limitations:

1. **Novel kernels:** While the Matérn family is a common default, it encodes assumptions of smoothness that are incompatible with the co-heritability search space. Alternative kernel choices may be able to capture greater structure in sharp aperiodic spaces.

2. **Non-GP approaches:** GP surrogates could be replaced with alternative models compatible with Bayesian optimization, such as random forests or Bayesian neural networks. These may naturally represent sharp, non-smooth functions.

3. **Improving random search:** A simple random search was competitive despite lacking adaptation. Enhanced random search methods could further improve optimization in sharp spaces with minimal computation overhead. Additional heuristics like local search around promising samples may further improve random search performance.

By moving beyond standard Gaussian process models and benchmarks, there is an opportunity to develop Bayesian optimization techniques tailored to the characteristics of complex, real-world search problems like co-heritability optimization. Both model improvements and enhanced baselines warrant investigation to better equip researchers to tackle searching sharp and aperiodic function spaces.

## Acknowledgments and Disclosure of Funding

This work is partially supported by NSF III 2046795, NSF IIS 2205329, NIH 1R01MH116226-01A, NIFA award 2020-67021-32799, and Google Inc.

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

## Appendix A    Run Times

Table 1: The mean and standard deviation of time, in seconds, needed to run each search experiment.

| Function | BO-EI | BO-UCB | Random | Iteration |
|---|---|---|---|---|
| Gaussian($\sigma = 0.1$) | $7.110 \pm 1.01$ | $5.999 \pm 0.90$ | $0.002 \pm 0.0$ | 14 |
| Gaussian($\sigma = 0.02$) | $7.460 \pm 1.43$ | $21.933 \pm 5.01$ | $0.005 \pm 0.0$ | 30 |
| Gaussian($\sigma = 0.005$) | $12.18 \pm 4.99$ | $17.517 \pm 4.89$ | $0.004 \pm 0.0$ | 30 |
| Michalewicz (2D) | $20.464 \pm 13.38$ | $62.303 \pm 36.83$ | $0.004 \pm 0.0$ | 40 |
| Eggholder | $107.74 \pm 78.39$ | $393.82 \pm 231.541$ | $0.013 \pm 0.0$ | 80 |
| Co-Heritability (for Narea) | $149.494 \pm 36.39$ | $1053.221 \pm 137.76$ | $0.0141 \pm 0.0$ | 310 |
| Co-Heritability (for SLA) | $160.060 \pm 39.39$ | $1657.262 \pm 511.47$ | $0.0171 \pm 0.0$ | 310 |

## Appendix B    Linear Mixed Effect Models (LLM)

Linear mixed effect models are characterized by the following hierarchical conditional distribution:

$$(Y | U = u) \sim \mathcal{N}(\mathbf{Z}\mathbf{\Lambda}_\theta u + \mathbf{X}\beta, \sigma^2 \mathbf{I}_n) \tag{1}$$

The conditional distribution can be broken down into modeling two types of variables 1) fixed effects 2) random effects. The data from the fixed effects is represented in a $n \times p$ matrix, $\mathbf{X}$, and random effect matrix is represented in a $n \times q$ matrix, $\mathbf{Z}$.

The random variable $U$ is called the spherical random effect. This creates a spherical probability density around the random effect:

$$U \sim \mathcal{N}(0, \sigma^2 \mathbf{I}_q) \tag{2}$$

The model parameters for a linear mixed effect model are $\beta, \sigma, \theta$:

- $\beta$ is p-dimensional weight vector where each weight corresponds to a fixed effect
- $\theta$ is a m-dimensional vector with covariance information for each random effects
- $\sigma$ is a shared scaling value for the variance

### B.1    Calculating Co-Heritability

Heritability ($h^2$) quantifies the degree to which variation in the presence of a trait within a population can be attributed to genetic differences versus environmental factors. Although there are several ways to calculate heritability, one such method is shown in Equation 3.

$$h^2 = \frac{\sigma_g}{\sigma_g + \sigma_e + \sigma_r} \tag{3}$$

In this equation $\sigma_g$ is the genetic variance, $\sigma_e$ is the environmental variance and $\sigma_r$ is the residual variance. Each of these variances are found in the covariance matrix ($\Lambda_\theta$) after filling LLM on trait, genetic and environmental factors for a set of plants.

Co-Heritability ($coh^2$) between two traits ($t_1, t_2$), can be calculated by combining the heritability of each trait and the covariance between the traits.

$$coh^2 = h^2_{t_1} h^2_{t_2} cov(t_1, t_2) \tag{4}$$

