# OpenReview forum: "Rethinking Bayesian Optimization with Gaussian Processes: Insights from Hyperspectral Trait Search"
_NeurIPS.cc/2023/Workshop/AI4Science — NeurIPS2023-AI4Science Poster_

### Official Review · Reviewer_jwoH · 2023-10-23
**Limitation analysis of standard Gaussian process Bayesian optimisation for sharp non-smooth and aperiodic hypothesis spaces**

**Rating:** 5
**Confidence:** 4

**Review:**

This is an attention paper that analyses the limitation of standard Gaussian processes (GP) regression for Bayesian optimisation (BO) in the case of co-heritability search from hyperspectral reflectance data, having sharp and aperiodic hypothesis space. In addition, toy data and two complex mathematical functions from evolutionary algorithm benchmarking are further utilised to support the findings. Paper is clearly written and structured. Originality of the paper is on showing a scientific optimisation problem where conventional GP assumptions does not hold. It gives practitioners a useful knowledge of when not relying on the out-of-box BO-GP. It is quite well-known that standards GP results smooth solutions, and in that sense paper proposes a new benchmark (which should be published) for developing better solutions. Paper itself has a limited discussion (some general level techniques listed) or aims towards developing novel approach to tackle the limitations, though.

Pros
- Showing negative result and issues of applying plain GP for BO
- Real-world challenging scientific problem and data
- Empirically shown results and comparison (accuracy and comp. time) with random search

Cons
- Missing alternative solutions (expect random search) for the problem; some future work listed, though
- Limited novelty in algorithms
- Lack of information about publishing the new benchmark dataset for BO as an open dataset

---

### Official Review · Reviewer_YocE · 2023-10-24
**Good empirical work, but conclusions about Gaussian processes are wrong and overly broad**

**Rating:** 5
**Confidence:** 4

**Review:**

This paper documents a case study of Bayesian optimization with Gaussian processes where GP-BO fails to outperform random search. The input feature space is spectral readings from plants, and the output space is a metric of heritability.

I think the main strength of the study is its empirical study of how BO can underperform random search for misspecified models (e.g. aperiodic functions with short lengthscales with the Matern kernel). The study is well-conducted and yields some useful insights for models selection.

My main issue with the paper is that all the conclusions are wrong/misstated. For example, the authors conclude that:

> Our results highlight fundamental limitations of Gaussian processes for sharp, aperiodic functions like those found in co-heritability search spaces.

This is not true: Gaussian processes actually have no fundamental limitations for sharp or aperiodic functions. _This is instead a limitation of the Matern family of kernels (or more generally a limitation of stationary smooth kernels)._ A Gaussian process with a different kernel would not necessarily have this limitation: for example, the "white noise" kernel $k(x,y)=\delta(x-y)$ makes no assumption of smoothness or periodicity. I think the conclusions could easily be corrected by restricting the claims about "Gaussian processes" to only "Gaussian processes with the Matern kernel" or "Gaussian processes with standard kernels".

Furthermore, assuming too much smoothness is known to cause issues in BO: see for example [1]. If this paper is to be extended to a full conference submission, I think the authors should engage with previous literature on this.

Finally, one important aspect in the study which the authors seem to overlook is the effect of hyperparameters: both GP hyperparameters and the $\beta$ hyperparameter in UCB. These critically affect performance, in particular how similar GP-BO behaves to random search. For example:

- As $\beta\to\infty$, GP-BO with UCB will behave more similarly to random search (effectively only choosing uncertain points). It is clear that in this extreme there will be little difference between GP-BO and random search.
- As the lengthscale $\ell\to0$ GP-BO will also behave like random search (because the covariance between data points will grow smaller).
- With low noise and kernel amplitude, GP-BO with EI will effectively stop exploring because $P(y > y_{best})$ becomes vanishingly small for points away from $x_{best}$.

Knowing this, it is clear to me that the phenomena observed by the authors will definitely be influenced by the hyperparameters, yet the authors do not study this. Instead, they state "We further note that any additional hyperparameter search adds additional computational overhead – despite only limited gains in performance". I doubt this statement is true. Perhaps the authors were just not tuning in the right range?

Overall, I think this paper should be re-presented acknowledging that GP-BO outperforming random search is clearly contingent on the kernel choice and hyperparameters, then presenting their analysis, and ideally studying the effect of hyperparameters. The main conclusion could be changed to "choice of kernel and hyperparameters are important".

One other note: the authors state that the cost of fitting the GP is $O(N^3)$ at each step. However, if the kernel hyperparameters are not changed then the cost should actually be $O(N^2)$ with a "proper" implementation which caches the Cholesky decomposition of the kernel matrix and performs a rank-1 update to include the new data point at a cost of $O(N^2)$. Even if the authors' implementation is a "lazy" one which re-computes the kernel matrix and its Cholesky decomposition in $O(N^3)$ time, it is inappropriate to suggest that this is a fundamental property of GP-BO.

Overall, considering what I see as many errors in the manuscript I will give a score of 5, but I am happy to see the paper accepted if the errors are corrected (which should be easy).

[1] Mathematical nuances of Gaussian process-driven autonomous experimentation